# Does the Lightning Process Training Programme Reduce Chronic Fatigue in Adolescent and Young Adult Cancer Survivors? A Mixed-Methods Pilot Study

**DOI:** 10.3390/cancers13164076

**Published:** 2021-08-13

**Authors:** Lena Fauske, Øyvind S. Bruland, Alv A. Dahl, Aase Myklebostad, Silje E. Reme

**Affiliations:** 1Department of Oncology, Norwegian Radium Hospital, Oslo University Hospital, 0379 Oslo, Norway; osb@ous-hf.no (Ø.S.B.); a.a.dahl@medisin.uio.no (A.A.D.); amk@ous-hf.no (A.M.); 2Department of Interdisciplinary Health Sciences, Institute of Health and Society, University of Oslo, 0317 Oslo, Norway; 3Institute of Clinical Medicine, University of Oslo, 0318 Oslo, Norway; 4Department of Psychology, Faculty of Social Sciences, University of Oslo, 0317 Oslo, Norway; s.e.reme@psykologi.uio.no; 5Department of Pain Management and Research, Oslo University Hospital, 0318 Oslo, Norway

**Keywords:** cancer-related fatigue, sarcoma, lymphoma, quality of life, qualitative research, cognitive training program, predictive processing, integrative cancer care

## Abstract

**Simple Summary:**

Chronic fatigue is a common late adverse effect following oncological therapies. No effective treatments exist, although cognitive behaviour therapy has been reported to offer some benefits. The Lightning Process^®^ is a three-day educational training programme with a six-month follow-up comprising elements derived from cognitive behaviour therapy, neurolinguistic programming, and stress theory, which are presented in a condensed form. This pilot intervention study represents the first systematic account of the experience and perceived efficacy of the LP training programme in 13 adolescent and young adult cancer survivors treated for sarcoma or Hodgkin lymphoma. Statistically significant improvements were documented for all the patient-reported outcome measure (PROM) questionnaires comparing the pre- and post-intervention periods. The qualitative findings of the interviews corresponded well with the PROMs findings, as participants emphasised that they now experienced both less fatigue and explicit improvement in their energy level.

**Abstract:**

Background: We report on a pilot intervention study exploring the efficacy of the Lightning Process^®^ training programme for reducing chronic fatigue and improving health-related quality of life in cancer survivors. Methods: 13 adolescent and young adult cancer survivors previously treated for sarcoma or Hodgkin lymphoma were enrolled. A mixed-methods approach was applied. This involved the use of five validated patient-reported outcome measure (PROM) questionnaires at baseline and the three- and six-month follow-up points to obtain quantitative data. Semi-structured interviews were conducted after the intervention with emphasis on the participants’ experiences and outcomes. A reflexive thematic analysis was applied to the transcripts. Results: A significant reduction (*p* < 0.001) in the total fatigue score from baseline to the three- and six-month follow-up points was documented. The correlation coefficients between the various PROMs at baseline and the six-month follow-up point indicated considerable overlap between the measures. The qualitative findings of the interviews corresponded well with the PROM findings. Most participants experienced both less fatigue and explicit improvement in their energy level. The aspects of the intervention found to be particularly helpful were the theoretical rationale and the coping techniques mediated. Conclusion: These encouraging results here reported should be of interest to the general oncological community, although they require confirmation through a larger and controlled study.

## 1. Introduction

As a well-established adverse effect following oncological treatment, chronic fatigue (duration ≥ six months) represents one of the most common challenges faced by cancer survivors [1,2]. Fatigue in patients with cancer has been underreported, underdiagnosed, and undertreated. Fatigue is defined as a distressing, persistent, subjective sense of physical, emotional, and/or cognitive tiredness or exhaustion related to cancer or cancer treatment not proportional to recent activity [2]. Although fatigue usually improves in the first year following treatment completion, 25–30% of patients continue to experience fatigue for years that may persist for up to 5 years or even longer after completion of treatment. Fatigue has a negative impact on work, family and social life, and it leads to significant impairments in overall quality of life [1]. In terms of tackling chronic fatigue among the general population, cognitive behaviour therapy (CBT) has been documented to offer certain benefits, albeit with only small to moderate effect sizes [3,4]. However, due to the low number of high-quality studies conducted so far, two recent reviews have concluded that no psychological or integrative interventions for addressing cancer-related fatigue can be recommended with confidence [5,6].

Adolescent and young adult (AYA) cancer survivors are defined as individuals who were aged between 15 and 39 years at the time of their initial cancer diagnosis [7]. AYAs are known to experience several challenges after undergoing cancer treatment, and they require age-appropriate and flexible care-related education to help them cope with long-term survivorship [8]. Yet, the number of prospective intervention studies designed to relieve the distress and improve the psychosocial wellbeing of AYA cancer survivors remains low, which indicates that more studies are needed [9,10].

The present study investigated the experience and efficacy of Phil Parker’s Lightning Process^®^ (LP), a three-day intervention programme claimed to be helpful in relation to various conditions, including myalgic encephalomyelitis/chronic fatigue syndrome (ME/CFS). Unlike patients with ME/CFS, our participants had an oncological history, which most probably explained their fatigue. However, when fatigue persists for years in AYA cancer survivors without any medical comorbidities, it is important to test new treatment approaches. The LP intervention investigated in this study represents such an approach.

The LP is a non-medical training programme that combines concepts derived from the fields of neuro-linguistic programming, positive psychology and self-coaching [11]. A key assumption of the LP is that chronic fatigue arises from dysregulations of the central and autonomic nervous systems, thereby resulting in a “false alarm” that can be turned off through top-down mental processes [11]. To the best of our knowledge, this is the first study to investigate the efficacy of the LP in relation to AYA cancer survivors with chronic fatigue.

In this study, we sought to explore whether an intervention involving the LP could significantly reduce the level of fatigue and enhance the health-related quality of life (HR-QoL) of AYAs who have been treated for sarcomas or lymphomas. Based on the documented effects of similar approaches to chronic fatigue [3,5], as well as on promising evidence concerning the effects of the LP [12], we hypothesised that the intervention would result in reduced levels of fatigue and enhanced HR-QoL, as reflected in the participants’ questionnaire responses. In addition, we hypothesised that qualitative interviews would provide valuable information to supplement the data derived from the patient-reported outcome measures (PROMs) and also reveal heterogeneity with regard to the participants’ experiences, as suggested by a previous qualitative study of the LP [13].

## 2. Materials and Methods

The present study was a non-blinded pre–post intervention study involving a mixed-methods approach. Several validated PROMs were used at baseline as well as during the follow-up period. After the completion of the full six-month follow-up period, semi-structured interviews were conducted.

### 2.1. Patient Demographics and Clinical Information

The participating patients were previously treated for sarcoma or Hodgkin lymphoma at the Norwegian Radium Hospital, Oslo University Hospital (NRH-OUH). The participants were all undergoing regular long-term clinical follow-up at the two diagnosis-specific outpatient clinics at the NRH-OUH, and they were all assumed to be cured of their malignancy (Table 1). The symptoms of chronic fatigue were present for years and repeatedly been documented in the participants’ clinical records as being their main late adverse effect following treatment and severely impairing their physical and psychosocial functioning. The well-known medical causes of fatigue (e.g., endocrinological, post-infectious or mental disorders) were ruled out. One reason for selecting survivors with the two diagnoses was the standard recommendation of at least 10 years follow up at the NRH-OUH.

One oncologist and an oncological nurse invited 17 AYA cancer survivors to take part in the study. Thirteen of those invited consented to participate. Exclusion criteria were mental disorders and the inability to speak and write Norwegian. Two patients declined, whereas two patients were unable to participate due to logistical reasons (Figure 1). Among the 13 AYA cancer survivors who agreed to participate in this study, ten were female and three were male. Their median age was 30 years (range: 21–36 years). The participants’ baseline demographic and clinical data are presented in Table 1. Two participants were diagnosed with osteosarcoma, one with Ewing sarcoma, one with chondrosarcoma, two with soft tissue sarcoma, and seven with Hodgkin lymphoma. One of the participants diagnosed with osteosarcoma had undergone a pulmectomy. The extent of all the participants’ oncological therapies was substantial, including high-dose chemotherapy with stem cell support in four of the lymphoma patients (Table 2). The participant diagnosed with chondrosarcoma, who was treated with extensive surgery only, had experienced considerable and long-term postoperative complications.

### 2.2. Study Procedures

Each participant received a letter providing detailed information regarding all the relevant aspects of the study and inviting them to provide written informed consent to participate. The letter stated that declining to take part would not affect their further treatment and/or follow up. The information also stated that they were free to withdraw from the study at any point without consequences. Usually, the LP course is paid for privately, but in this study, it was provided free of charge both for the participants and the hospital. The data protection officer at the NRH-OUH approved the study (approval number #19/00495).

#### 2.2.1. Quantitative Study

Self-reported questionnaires with established psychometric properties covering several PROMs were completed by the participants at baseline, one week after the LP course, and then during their three- and six-month follow-up appointments (Figure 1). More specifically, the total fatigue score was self-rated by the Fatigue Questionnaire (FQ) [14], total depression score by the Patient Health Questionnaire 9 (PHQ-9) [15], work and social adjustment score by the Work and Social Adjustment Scale (WSAS) [16], score of patient satisfaction by the Client Satisfaction 8 (CSQ-8) [17] and their HR-QoL scores by the Short-Form Health Survey 36 (SF-36) [18,19]. The CSQ-8 was completed one week after the participants finished the LP course and again at the six-month follow-up point (Figure 1). In the CSQ-8, the total score ranged from 8–32, with higher scores indicating a higher degree of satisfaction.

#### 2.2.2. Statistical Analyses

Changes in the mean scores from baseline to the three- and six-month follow-up points were analysed by means of paired sample *t*-tests. Any changes in the mean scores for the PROMs with 95% confidence intervals and standard deviations (SDs) from baseline to the three- and six-month follow-up points, as well as between the follow-up sessions, were considered. Changes between the measurement points of ≥0.5 SD from the baseline mean scores for the PROMs were clinically significant [20]. Spearman’s rho was used to analyse the correlations between the various PROMs, including the degrees of explained variance and overlap. The explained variance being the squared correlation coefficient in per cent. The statistical software used for the analyses was SPSS version 25 for PC (IBM Corporation, Armonk, NY, USA).

#### 2.2.3. Qualitative Study

The qualitative part of this study applied a phenomenological and hermeneutical approach to investigate the participants’ individual experiences (phenomena) as they manifest in both daily life and specific situations. This approach provided a psychosocial perspective on the participants’ concepts of health and illness [21].

The interviews were conducted either at the NRH-OUH (*n* = 8) or, when restrictions related to the COVID-19 pandemic necessitated, by telephone (*n* = 3). L.F. (*n* = 7) and A.M. (*n* = 4) conducted all the interviews after the patients had finished their six-month follow-up sessions. The interview guide contained only open-ended questions. The participants were first invited to narrate a short version of their medical history with an emphasis on daily life after treatment. This gave us an insight into their challenges before the intervention. They were further asked how they experienced the LP course and follow-up. What did they find useful and important or not, whether they used the technique today and how they envisioned the future use of the technique? They were also interviewed about how the LP programme had affected them in their everyday lives regarding family life, social life, physical activity/leisure and vocational life.

The interviews had a median duration of 32 minutes (range: 19–55 min). The semi-structured interviews were transcribed verbatim by a medical secretary. The participants are named P1–13 in Table 3. Participants 8 and 13 withdrew from the study shortly after the LP course.

#### 2.2.4. Analysis

A reflexive thematic analysis approach [22,23] was used to identify the patterns of meaning across the dataset related to the research questions being addressed. The patterns were identified through a rigorous process of data familiarisation, data coding, theme development and revision. The complete interview transcripts were inductively coded in detail (i.e., inclusive and extensive) by hand, principally by L.F. and partly by A.M., both unique and overlapping. Then, the codes were divided into categories and themes [22]. Throughout the process of analysis, the coders regularly returned to the original transcripts to check the themes and quotes, as well as to ensure that the meaning had not been lost during either the interpretation or the translation [24].

### 2.3. Description of the Intervention

Prior to attending the three-day LP course, the participants had a telephone conversation with the course instructor to clarify any issues they might have and to allow the instructor to assess whether they were sufficiently motivated to proceed with the intervention. The latter was based on a standard LP checklist. The LP course, primarily developed for patients with chronic fatigue and pain, and not for AYA cancer survivors was delivered in three consecutive half-day seminars (four to five hours each), which were attended by groups of three to six participants. Each seminar included both a theory session comprising psychoeducation regarding stress physiology, mind and body interaction and chronic fatigue, and helpful or unhelpful thought processes and a practical session to put the learned skills into practice [11,13]. The participants were trained to recognise their thoughts and symptoms, and they were taught how to influence and avoid unhelpful physiological responses. The participants were encouraged to begin practising the new techniques immediately. Upon completion of the course, a monthly follow-up telephone session with the LP course instructor was provided for a period of six months to support the participants in adopting the new coping skills.

The LP course was delivered by Kristin Blaker, a certified LP instructor from the Phil Parker Training Institute. She worked without any economic compensation for the purpose of this study. She did not have access to either the participants’ medical records or the dataset. In addition, she did not take part in the analysis and evaluation of the obtained data.

## 3. Results

The median time from diagnosis to the intervention was seven years, and the median time since receiving the final round of chemotherapy was six years (Table 1).

### 3.1. Quantitative Study

Eleven of the 13 participants completed all PROMs at all time points. The participants’ individual scores are provided in Appendix A. A significant reduction in the participants’ total FQ scores from baseline to the three-month follow-up point was observed (*p* < 0.001), with nine participants maintaining that reduction (*p* < 0.001) at the six-month follow-up point (Figure 2A and Appendix A). Any further improvements in terms of the score changes from the three- to the six-month follow-up point were non-significant. A reduction in fatigue was also observed in the remaining three participants, although these changes were not significant (Appendix A). Among the nine patients who exhibited significant improvements at the six-month follow-up point, the median total FQ score reduction was 15 (Range: 5–26).

The changes in the PHQ-9, the WSAS, and the SF-36 mental composite scale (MCS) scores from baseline are displayed in Figure 2B–D and Appendix A. In terms of the participants’ PHQ-9 and WSAS scores, as with the total FQ scores, similar and significant positive changes (*p* < 0.001) were reported over time. Again, the observed changes had already occurred by the three-month follow-up point. Moreover, the changes were maintained at the six-month follow-up point, albeit without significant changes occurring between the three- and six-month follow-up points (Appendix A). Regarding the SF-36 scores, the findings concerning the MCS are reported in Figure 2C and Appendix A, with improvements from mean score 35 at baseline, to 50 at three months and 45 at six months. The analysis concerning the physical composite scale (PCS) of the SF-36 is shown in Appendix A, but no significant improvement was documented.

One week after finishing the LP course, the participants rated their experiences related to satisfaction in the CSQ-8 with a mean score of 25.7 (SD 4.3). At the six-month follow-up, the mean score remained practically unchanged at 26.6 (SD 5.2). This high degree of satisfaction is encouraging, taking into consideration that the maximum score of the CSQ-8 is 32.

We estimated the correlation coefficients between the various PROMs, both at baseline and at the six-month follow-up point (Appendix A). A moderate overlap between the participants’ PHQ-9 and FQ scores was observed at baseline, with 33% of the data variation in the former being explained by the variation of the latter. Furthermore, at the six-month follow-up point, the participants’ PHQ-9 and WSAS scores were found to be highly correlated, with an explained variance of 58%. This implied that these PROMs to a large degree measured the same phenomena. The PHQ-9 and FQ scores at six months revealed 64% explained variance, while a calculation comparing the FQ and CSQ-8 scores revealed 52% explained variance. Hence, we concluded that the PROMs overlapped considerably.

### 3.2. Qualitative Study

Seven main themes were identified during the analysis: a feeling of more energy and “joie de vivre”; changing dysfunctional thought patterns using a well-learned process; integrating the technique into daily life and using the process as needed; daring to challenge oneself and move outside one’s comfort zone; better life with family and friends and increased energy at work; importance of follow-up dialogues to maintain the process/improvements; feeling that the course/intervention was good but I do not think the process suited me.

The 11 participants that took part in the interviews all reported a clear improvement in their fatigue levels. Indeed, they reported feeling either much better or very much better regarding their energy level. Moreover, they stated that the energy-related improvement could be attributed to the LP course. They commented on how the intervention had challenged them to recognise and address what caused them to feel stress and reduce their energy level, and they related how they were encouraged to use the same process to reverse any destructive thoughts and negative patterns of action. As expressed by P2, “What I benefited most from was noticing how I responded to stress […] after the course, I completely changed the way I thought about my physical condition”. The participants distinguished between using all the components of the LP process regularly and using just the simple mental techniques intended to stop destructive thoughts. All 11 participants reported that they still used the LP techniques they had learned or, at least, parts of the techniques in one form or another. They did so especially when life became challenging or when they felt that negative thoughts were prevailing. Six participants reported that they had integrated all the LP techniques into their lives and so used the whole process when required. As emphasized by P11, “When I feel I am in an awful state, I do the whole process, like some days I do it many times and sometimes I do it a couple of times a week, you see. It depends on how my life is at the time”. They stated that they had become more aware of situations that triggered and affected their fatigue experiences, and they considered that they now had a useful tool for influencing both the extent and experience of such symptoms.

In general, the participants reported improved quality of life with more social and physical activity than they had prior to the intervention. Further, they now had a better family life, and some also felt more energetic at work. The latter phenomenon was reflected in increased workplace attendance by some participants, especially by P4, “I felt that I could work a bit more efficiently and I had a bit more energy to do things after work that were very important to me. This June I started a full-time job in another company (compared to part-time, 30%, before the course), which would have been impossible without the Lightning Process […]. Now I am financially independent of my parents, which has been my dream for about ten years”. In particular, the participants reported deriving motivation from the fact that they now felt better able to challenge themselves and to move outside their comfort zone without being afraid of becoming too fatigued.

The follow-up period was mentioned by most participants as being essential for the positive changes to persist over time. As stated by participant 1, “The most important thing has been the follow-up. The phone calls. One-on-one dialogues, definitely […]. I think it has been useful to have those talks regularly so that you keep going with the process and what you have learned”. Finally, all 11 participants confirmed that the intervention had not worsened their health or caused them any negative side effects. Table 3 presents selected quotations from the participants.

## 4. Discussion

Despite the low number of participants included in this study, statistically significant improvements were documented for all the PROMs from the pre- to the post-intervention period. Although the identified improvements were maintained, no further benefits were observed from the three- to the six-month follow-up point. Still, the improvements observed from baseline to the six-month follow-up point were significant. The reductions in the participants’ total fatigue scores were remarkable, since no changes in their overall level of fatigue, as subjectively expressed by the participants themselves, were reported over the preceding years. Improvements were documented in terms of significant reductions in the mean group scores from baseline to the three-month follow-up point, which supported our hypothesis.

The quantitative findings corresponded well with the findings of the interviews. Here, the participants emphasised that they now experienced both less fatigue and a clear improvement in their energy level. The aspects of the intervention that they found to be particularly helpful were the theoretical rationale and the coping techniques that they learned and practised. However, some heterogeneity was found regarding both how they used the LP techniques and how much they had integrated those techniques into their lives.

As mentioned above, within the LP paradigm, chronic fatigue is suggested to be maintained by dysregulations of the autonomic and central nervous systems. This rationale corresponds well with the sustained arousal model of chronic fatigue syndrome [25] and the more general cognitive activation theory of stress [26], which both suggest causal links between sustained arousal and the experience of fatigue. During the LP course, these theoretical perspectives are explicitly explained to participants to validate their symptoms as being real, rather than “imagined”. Further, although such symptoms can be understood through plausible neurophysiological mechanisms, the LP stresses that they can still be modified through mental processes, which is what the participants learned during the seminars [11].

The LP approach is neither purely psychological nor purely biological. Rather, it aims to integrate both perspectives to understand the vicious cycles that maintain both fatigue and disability. The participants in our study emphasised that the intervention had caused them to gain new awareness and knowledge about how negative thoughts and worries are stressful and thereby consume a lot of energy. Moreover, they learned that they were unconsciously maintaining the destructive patterns that contributed to the fatigue. Several participants noted that they were now able to recognise the warning signs and, further, that they were provided with a tool to help prevent the thoughts and actions associated with stress and the corresponding reduction in their energy level. They expressed being able to do more after completing the course, including being more socially and physically active. Some participants were even able to increase their workplace participation. For many participants, the fact that they now managed to go outside their comfort zone was a great motivation for continuing to implement the LP techniques.

The positive effects of the LP observed in this study can also be understood within the recently developed predictive processing framework. This emerging approach to understanding mind–body connections arose from neuroscientific advances demonstrating how the brain processes interoceptive inputs and makes top-down predictions about expected sensory inputs [27,28]. Put simply, the brain acts as a kind of prediction machine that matches incoming data with prior predictions and, if there is a mismatch (prediction error), updates the predictions accordingly. However, in some cases, the prior predictions are so strong, partly due to the prior learning history or contextual cues, that the sensory input is overruled [27,29], similar to what occurs in relation to placebo and nocebo effects [30]. In patients with cancer-related chronic fatigue, the previous history of a tough disease and intensive treatments could lead to strong perceptual predictions with very high previous likelihoods. These prior predictions could then influence the individual patient’s experience to such a degree that sensory information (or the lack thereof) is not assigned priority, thereby resulting in the continuation of fatigue-related symptoms.

In light of this new theoretical approach, the LP could be considered to work through both top-down psychic mechanisms (i.e., by strengthening the individual’s belief in their ability to control their symptoms and so updating prior predictions) and bottom-up mechanisms (i.e., by facilitating new learning through behavioural experiments that provide positive sensory inputs that contribute to updating prior predictions).

During the interviews, all the participants revealed how they were either satisfied or very satisfied with the intervention. However, one mismatch did occur, as one participant reported consistently low scores on the PROMs, while during the interview, she expressed satisfaction with the course and felt she had benefited from it, although she did not feel comfortable with all aspects of the process. We also observed that the two participants who chose to withdraw from the study had most recently received their diagnosis (24 and 30 months, respectively). Despite the low number of participants, this finding raises the question of whether the short time since their oncological diagnosis partly explains their withdrawal from the intervention.

In contrast to a prior qualitative study involving the LP, in which two out of nine young ME/CFS patients reported a lack of improvement in their energy level and a resultant feeling of being blamed [13], none of our participants felt that the intervention implied negative experiences. Several participants reported that the long follow-up period after the intervention (i.e., six months) and the frequent contact with the course leader were important to them in terms of implementing the techniques and maintaining the useful changes.

An obvious limitation of our study concerns the lack of a control group. It should also be mentioned that our study recruited ten females but only three males. We believe that the improvements described in the interviews and reflected in the PROMs are less likely due to chance or change in life events and more likely attributable to the intervention. Yet, “finally being taken seriously”, receiving considerable attention, and being invited to participate in a clinical study seeking to reduce fatigue cannot be ruled out as partial explanations. As this is the first study to investigate the efficacy of the LP among AYA cancer survivors, our results cannot be compared with those of other studies.

## 5. Conclusions

Chronic fatigue is a common and disabling late effect from cancer in which few effective interventions exist. The positive results reported here are therefore encouraging and suggest the need for controlled and ideally randomised studies among larger cohorts of cancer survivors who are struggling with chronic fatigue.

## Figures and Tables

**Figure 1 cancers-13-04076-f001:**
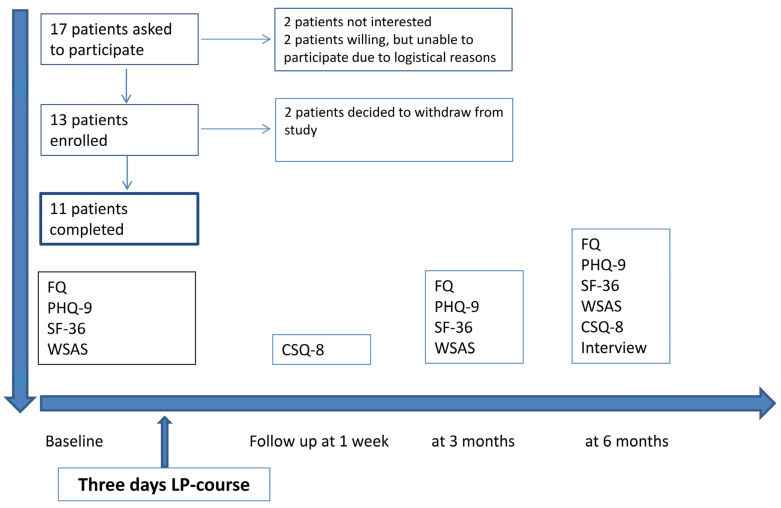
Flow Chart Patients and Study Outline. Adherence to study and timeline related to intervention, patient-reported outcome measures (PROMs) and interview. Fatigue Questionnaire (FQ), Patient Health Questionnaire 9 (PHQ-9), Short-Form Health Survey 36 (SF-36), Work and Social Adjustment Scale (WSAS) and Client Satisfaction 8 (CSQ-8).

**Figure 2 cancers-13-04076-f002:**
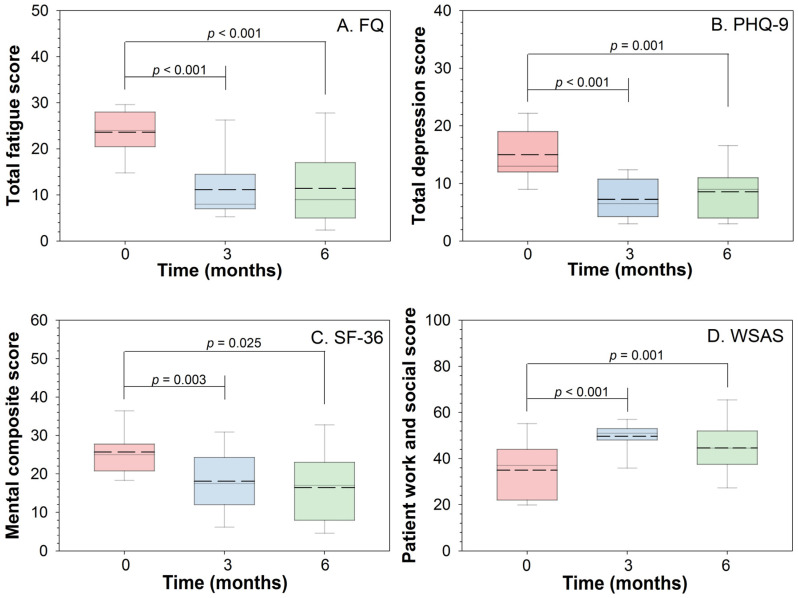
Patient-reported outcome measures. (**A**) Fatigue Questionnaire (FQ); (**B**) Patient Health Questionnaire 9 (PHQ-9); (**C**) Short-Form Health Survey 36 (SF-36); (**D**) Work and Social Adjustment Scale (WSAS). The bottom of the box represents the 25th percentile, the dashed line represents the mean, the solid line represents the median, the top of the box represents the 75th percentile, and the whiskers represent the 5th and 95th percentiles.

**Table 1 cancers-13-04076-t001:** Baseline demographic and clinical data.

Demographics and Diagnosis	Numbers
**Total number of patients**	13
**Median age** (years and range)	30 (21–36)
**Sex**	
Female	10
Male	3
**Relationship status**	
Married	1
Co-habitant	7
Single	5
Children	4
**Time since primary diagnosis** (median in years and range)	7 (2–12)
**Time since receiving last systemic therapy** (median in years and range)	6 (1–11)
**Diagnosis**	
**Bone sarcomas**	**4**
Osteosarcoma	2
Ewing sarcoma	1
Chondrosarcoma	1
**Soft tissue sarcoma**	**2**
Rhabdomyosarcoma	1
Synovial sarcoma	1
**Hodgkin lymphoma**	**7**

**Table 2 cancers-13-04076-t002:** Oncological treatments.

Oncological Treatments	Number of Patients
**Sarcomas**	
**Surgery only**	**1**
**Surgery and adjuvant treatment**	**5**
**Radiotherapy**	**3**
**Chemotherapy**	**5**
Ewing sarcoma combo	2
Osteosarcoma combo	2
Doxorubicin/ifosfamide	1
**Number of chemotherapy combinations**	-
One line	3
Two lines	2
**Surgery for metastatic recurrence in lungs**	**2**
**Hodgkin lymphoma**	
**Radiotherapy**	**4**
**Chemotherapy**	**7**
**Number of conventional chemotherapy combinations**	-
One line	3
Two lines	3
Three lines or more	1
**High-dose chemotherapy with stem cell support**	**4**
Autologous	2
Allogeneic	2
**Immunotherapy**	**3**

**Table 3 cancers-13-04076-t003:** Selected quotations from the interviews reported by the 11 participants.

Themes	Selected Quotations
A feeling of more energy and “joie de vivre”	• Before the course I could sleep for four hours when I came home from work. Now I actually cannot remember the last time I slept after I got home. P1 • I felt much more exhausted before the course. I have got a lot more energy now […]. I feel things have gone much better since the course. […]. My body works better, I am less agitated, I have less pain and my energy level is a bit higher. And I can sleep better. P3 • After the course, I felt so positive. I had been given a new boost, I was happy in my daily life and I felt much more hopeful. P9 • The main thing for me, I think, was that I rediscovered my energy, that is where I have felt the most change. P6
Changing dysfunctional thought patterns using a well-learned process	• What I benefited most from was noticing how I responded to stress […] after the course, I completely changed the way I thought about my physical condition. P2 • We have been given different exercises to think about, when we are stuck in situations, for example, then I have done those exercises and trained my brain to think in a different way […]. I am more aware of the things I do and think—that everything is linked to feeling lethargic and tired. P5 • It is all about appreciating and thinking about all the good things in life instead of constantly focusing on what is not good, or what I did not manage to do, then you get into a vicious circle again. So, it has been helpful to become aware of that. P9
Has integrated the technique into daily life and uses the process as needed	• When I feel I am in an awful state, I do the whole process, like some days I do it many times and sometimes I do it a couple of times a week, you see. It depends on how my life is at the time. P11 • I have got a tool I can use for the rest of my life […]. I go through the process instead of resting and I start the rest of the day from there. P4 • I mostly use it in my head from day to day […], but if I have any negative feelings or thoughts […] I notice there is a greater effect if I use it physically. P12
Daring to challenge oneself and move outside one’s comfort zone	• I dare to do things a bit more without thinking so much about what sort of consequences they will have for me physically. It has helped me to be a bit more relaxed and then that maybe gives me a bit more energy. P2 • The most important progress has been to see that I can actually manage to do things. P9 • I find it a bit easier to trust that things will get better and that my body is healthy and things will be fine. [..]. After working two days a week, I can still get a visit on the Saturday and that does not make me feel worse. P11 • I now realize I cope better than I thought with stressful situations or situations that I find difficult mentally and I understand better what makes me depressed […]. I am now more robust, I would say. P2
A better life with family and friends and increased energy at work	• My family life is much better now I have got more energy and more stamina […]. Before the course I really was not sociable, and I was not up to much […] now things are better in every possible way. My partner would agree completely. P3 • I felt that I could work a bit more efficiently and I had a bit more energy to do things after work that were very important to me. This June I started a full-time job in another company (compared to part-time, 30%, before the course), which would have been impossible without the lightning process […]. Now I am financially independent of my parents, which has been my dream for about ten years. P4 • Before the course I did nothing, and I was hardly ever in social situations […]. Now I have the energy to be sociable and not be completely exhausted afterwards. (laughs) […]. I am really pleased about that. That was what I was hoping for, to start being with my family and friends again. P7 • The way I see it now, I feel that I can cope with daily life, I can cope with life as a mother, I can function properly, and I can get things done. I take the initiative to do things, which was a good bit more difficult before. P9
Importance of follow-up dialogues to maintain the process/improvements	• The most important thing has been the follow-up. The phone calls. One-on-one dialogues, definitely […]. I think it has been useful to have those talks regularly so that you keep going with the process and what you have learned. P1 • I really liked the follow-up afterwards, there were regular phone calls where we talked about how things were going and some homework for the next time. P7 • I think the most important thing was the period after the course where you were in touch with the instructor […] it was great to have someone you could hold on to. P4
Feeling that the course/intervention was good but I do not think the process suited me	• The process is not quite me […] I cannot keep up the daily exercises. P3 • In general, everything was very good, but the process itself was very strange to me […] mainly perhaps a bit outside my comfort zone. P6

## Data Availability

All data generated or analysed during this study are included in this published article (and its Appendix A). Statistical analysis code is available upon request. The dataset analysed during the present study cannot be shared under the current protocol and ethics committee approval.

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
