# Peer review of "Does the Lightning Process Training Programme Reduce Chronic Fatigue in Adolescent and Young Adult Cancer Survivors? A Mixed-Methods Pilot Study"

_cancers, 2021, doi:10.3390/cancers13164076_

Round 1
Reviewer 1 Report
Fauke et al. present an interesting article regarding a new system (the Lightning Process Training Program) to reduce chronic fatigue in AYAs survivors. This is definitely and important issue that AYAs struggle with and there is limited information regarding techniques or processes to assist these patients. This article can definitely add to the literature and provide a possible approach to help these patients. Agree that this would be strengthened with a control group, but this pilot study does provide some significant groundwork for later larger studies.
Some questions/concerns for the authors:
- Introduction:
- Lines 65-68: The authors state that “A key assumption of the LP is that chronic fatigue arises from dysregulations of the central and autonomic nervous system…” This is also repeated in the discussion portion of the paper. Is there any literature to back this assumption up? As this is a major foundation point to why you feel your program is going to work, your argument can be strengthened greatly by having literature that supports this argument. Basing an entire theory and therapy on an assumption would not feel like good medical practice. Please try to find support articles for this and reference them in the introduction and the discussion where this is discussed.
- Materials and Methods
- 1 Patient demographics and clinical information:
- I think that is good that the authors have documented the times from diagnosis and from completion of therapy, indicating that these patients are long term survivors. They have also documented that the chronic fatigue was in the chart several times. I would argue that it would be just as important to document how long each patient has had the diagnosis of chronic fatigue. Has it only been within 6 months, or several years? I feel that this important and may show that even patients with the diagnosis for several years can be helped.
- 3 Description of the intervention:
- Lines 168-173The authors discuss that each seminar had a theory session and a practical session. The theory session is described within the parentheses, why the practical session is separated in another new sentence. I would suggest rewording this section to make it clearer – it is difficult to understand that the authors are describing the practical session as it is in a new sentence.
- 1 Patient demographics and clinical information:
- Results:
- 1 Quantitative Study:
- Lines 186-189: The authors describe that nine of the 11 eleven participants had reduction at the six-month follow-up point. While this is important, I do think that it is just important to understand why the other 2 participants did not maintain reduction. This paper would benefit from adding this into the discussion portion of the paper.
- Lines 189-191: The authors document that they saw statistically significant changes from the baseline to three-month follow-up but not from the three month to 6 month time frame. This is discussed several times, but the authors do not give any reason why there was no difference at this time point. I feel the paper would benefit from an explanation of this data.
- 1 Quantitative Study:
- 2 Qualitative Study
- The information and quotes provided in the qualitative study is interesting and informative. I do think though, that it is important to discuss the questions that were asked to provide the reader with more information. For example, in lines 226-227, the authors report, “Moreover, they stated that the energy-related improvement could only be attributed to the LP course”. This is a very definitive statement, and without context of how the question was asked, it could be perceived that the it was asked in such a way that this answer was coerced. More background would help know that this type of response was voluntary.
- Discussion
- Lines 330-332: The authors discuss that the two patients that withdrew from the study had most recently received their diagnosis. Again, I feel that reporting the specific times may be beneficial to understand when to present a program to patients.
- Minor grammatical considerations:
- Simple Summary:
- Line 16: “No effective treatment exist” should be “treatments”
- Line 17-18: “…with six months follow-up…” should be “month”
- Simple Summary:
- Line 60: “…more such studies” is an awkward phrase
- Lines 211-212: “…the mean score remained practically unchanged, standing 26.6.,” The word standing does not make sense here.
- Line 284: “As mention above…” should be “mentioned”
Author Response
Response to reviewer 1
We sincerely appreciate the favourable comments from the reviewer and the opportunity to submit a revised manuscript. We have, point by point, attempted to correct the manuscript accordingly.
Some questions/concerns for the authors:
Introduction:
- Lines 65-68: The authors state that “A key assumptionof the LP is that chronic fatigue arises from dysregulations of the central and autonomic nervous system…” This is also repeated in the discussion portion of the paper. Is there any literature to back this assumption up? As this is a major foundation point to why you feel your program is going to work, your argument can be strengthened greatly by having literature that supports this argument. Basing an entire theory and therapy on an assumption would not feel like good medical practice. Please try to find support articles for this and reference them in the introduction and the discussion where this is discussed.
We have now consulted key researchers in this field. They verified that the already referenced publication (Ref. 11) is the only one available specifically backing the assumption related to LP. The cited reference is actually a literature review where empirical support from both pre-clinical and clinical studies are provided as support for the key assumptions underlying LP. However, in the discussion part of our paper, we have cited three to four papers (Refs. 25, 26, 27, 28). They are all providing theoretical and/or empirical support for the key assumption mentioned above – but not specifically related to LP.
Materials and Methods
Patient demographics and clinical information:
- I think that is good that the authors have documented the times from diagnosis and from completion of therapy, indicating that these patients are long term survivors. They have also documented that the chronic fatigue was in the chart several times. I would argue that it would be just as important to document how long each patient has had the diagnosis of chronic fatigue. Has it only been within 6 months, or several years? I feel that this important and may show that even patients with the diagnosis for several years can be helped.
The fatigue had indeed been present for years. We have expanded the text in line 110 in the revised manuscript. The timeline is also presented in table 1.
Description of the intervention:
- Lines 168-173The authors discuss that each seminar had a theory session and a practical session. The theory session is described within the parentheses, why the practical session is separated in another new sentence. I would suggest rewording this section to make it clearer – it is difficult to understand that the authors are describing the practical session as it is in a new sentence.
We have now reworded this section and hope it is now clearer.
Results:
Quantitative Study:
- Lines 186-189: The authors describe that nine of the 11 eleven participants had reduction at the six-month follow-up point. While this is important, I do think that it is just important to understand why the other 2 participants did not maintain reduction. This paper would benefit from adding this into the discussion portion of the paper.
As part of the informed consent it is entirely up to each participant to withdraw from the study without any explanation. Hence, we have no data documented in this regard, but they orally conveyed shortly after the course that the LP-training program did not fit them.
- Lines 189-191: The authors document that they saw statistically significant changes from the baseline to three-month follow-up but not from the three month to 6 month time frame. This is discussed several times, but the authors do not give any reason why there was no difference at this time point. I feel the paper would benefit from an explanation of this data.
The major therapeutic changes took place between baseline and 3- month follow-up, and these changes were mainly consolidated between 3- and 6-month follow-up.
Qualitative Study:
- The information and quotes provided in the qualitative study is interesting and informative. I do think though, that it is important to discuss the questions that were asked to provide the reader with more information. For example, in lines 226-227, the authors report, “Moreover, they stated that the energy-related improvement could only be attributed to the LP course”. This is a very definitive statement, and without context of how the question was asked, it could be perceived that the it was asked in such a way that this answer was coerced. More background would help know that this type of response was voluntary.
We have deleted “only”.
The interview guide (currently in Norwegian) contains what we regard as open-ended questions. We have now added more information about this from line 178-185. If the editor/reviewers wish, we may translate the entire interview- guide to English and present as supplementary file.
Discussion
- Lines 330-332: The authors discuss that the two patients that withdrew from the study had most recently received their diagnosis. Again, I feel that reporting the specific times may be beneficial to understand when to present a program to patients.
We have now specified this in lines 400-402
Minor grammatical considerations:
- Simple Summary:
- Line 16: “No effective treatment exist” should be “treatments”
- Line 17-18: “…with six months follow-up…” should be “month”
- Main Text:
- Line 60: “…more such studies” is an awkward phrase
- Lines 211-212: “…the mean score remained practically unchanged, standing 26.6.,” The word standing does not make sense here.
- Line 284: “As mention above…” should be “mentioned”
We thank the reviewer for correcting grammar/linguistic errors - all issues have now been corrected accordingly.

Reviewer 2 Report
Overall, the authors present an interesting, novel and valuable paper evaluating the ‘Lightning Process’ training program aimed at improving chronic fatigue and HRQoL in young cancer survivors. There are some areas requiring clarification or minor suggested improvements to the presentation of the study.
Abstract:
Overall the abstract is a well written summary of the manuscript. There are some key details missing from the abstract however, which may have been presented in the “Simple summary” which should instead be focused on the background, aims and impact/importance of the research. I have highlighted these areas:
- The authors use cognitive [behavior or behavioural] therapy interchangeably when referring to CBT. Suggest using one term for consistency, preferrably the former.
- The abstract would benefit from a brief description of the Lightning Process program.
- Lines 31-32: It is not clear from the abstract what outcomes the semi-structured interviews assessed
- Line 32: Please include the number of survivors recruited for the study in the results (which should be moved from the second background sentence) and basic demographic information to characterise the sample (e.g. age, sex, time since diagnosis if available)
- Lines 33-34: “The correlation 33 coefficients between the various PROMs at baseline and the six-month follow-up point indicated 34 considerable overlap between the measures.” There is no detail provided about the “five validated patient-reported outcome measures” used, therefore these findings are not particularly valuable. It would be useful to briefly list them, or at least the overlapping outcome measures alluded to.
- Please add relevant statistics and/or p-values to the results.
- Lines 38-39: “These encouraging results here reported should be of interest to the general oncological community” Perhaps a more useful conclusion would be a summary of the key finding(s) or message or potential implications.
- It is not clear from the abstract that the program is designed for young cancer survivors. Please add to the background or to the methods (program description or eligibility criteria).
Introduction
The introduction offers a brief but adequate background of the topic, including engagement with relevant literature, to provide a good rationale for the study.
- The introduction is somewhat lacking in providing an understanding of the short/long term impact of (unaddressed) chronic fatigue in cancer survivors, and therefore the importance of programs like Lightning Process. Doing so would greatly strengthen the significance and rationale for the program and its evaluation.
- Line 63: Perhaps provide a brief definition of “myalgic encephalomyelitis” or more simply refer to it as “Chronic Fatigue Syndrome”
- “This study sought to explore whether an intervention involving the LP could significantly reduce the level of fatigue and enhance the health-related quality of life (HR-QoL) 71 of AYAs who have been treated for sarcomas or lymphomas.” HRQoL is not mentioned as an outcome in the abstract and may be worth including if indeed it is a primary focus.
Methods:
The methods are generally well described and appropriate for the research aims, yet lacking some details.
- Lines 86-87: Was the program designed specifically for Lymphoma and Sarcoma patients, or was there another rationale for only recruiting these patients and not survivors of other diagnoses? Could the program be relevant for other survivors or is further evaluation needed in these groups?
- Lines 87-89: The following sentence should be moved to the results “The median time from diagnosis to the intervention was seven years, and the median time since receiving the final round of chemotherapy was six years (Table 1).” Table 1 and Table 2 should also be moved to the results. Instead, additional information about participant eligibility is needed, including any exclusion criteria.
- Lines 92-93: “The symptoms of chronic fatigue had repeatedly been documented in the participants’ clinical records as being their main late adverse effect following” Given that fatigue was recorded in patient records as a problematic late effect, could they have been receiving other treatment(s) to manage their fatigue? Was this ruled out prior to participation, or noted, to account for the potential influence of other treatments which may have also improved fatigue and/or HRQoL?
- Lines 98-110: The description of the participants and their characteristics should appear at the start of the results, not in the methods.
- Line 122: Please define “NRH-OUH” in full.
- Lines 125-132: Additional information about the nature of the measures would be useful for the interpretation of the findings, i.e. possible response options and score ranges, scoring (e.g. higher score = more depression)
- Line 148: Please define “OUH-NRH” in full, if different to NRH-OUH
- Line 155: Additional information about the qualitative analysis is needed. Did LF and AM code unique transcripts, or an overlapping proportion? How was the coding tree developed and how were themes derived (in advance, or inductively)? What software (if any) was used to support analysis? What questions were asked in the interview and/or outcomes assessed?
- Lines 165-167: “and to allow the 165 instructor to assess whether they were sufficiently motivated to proceed with the intervention.” Without context, this may be interpreted as somewhat coercive. What measures were taken to ensure participants were ‘motivated’, or what criteria was used to determine if they were or were not?
Results:
Overall, the results are sound although could use some improvement in their presentation.
- Throughout the results, I suggest adding relevant statistics where results are described for completeness, for example the mean difference in scores, test statistics, and/or p-value.
- The readability/flow of the results would be enhanced by using the name of the outcome (e.g. fatigue, depression) rather than the acronyms of the formal measure names (e.g. FQ, PHQ), providing they are adequately described in the methods (as noted above). Alternatively, the results could benefit from additional subheadings for each outcome.
- Lines 189-191: “Any further improvements in terms of the….” This sentence isn’t clear. Perhaps reword to indicate more clearly that a reduction in fatigue was also observed in the remaining three patients, although these changes were not significant.
- Lines 205-206: A brief summary of the PCS findings are needed, rather than simply stating “The analysis concerning the PCS is shown in Supplemental Table 5.”
- Lines 207-208: Please move the following to the methods: “The CSQ-8 was completed one week after the participants finished the LP course and 207 again at the six-month follow-up point…higher scores indicat[ed] a higher degree of satisfaction.” It would also be useful to know what the total possible score is, to understand the scores better.
- Lines 214-220: “A similar calculation comparing the participants’ PHQ-9 and FQ score…” This is vague. How is it different to the data presented a few sentences prior “A moderate overlap 214 between the participants’ PHQ-9 and FQ scores was observed at baseline…”.
- Lines 221-222: “Hence, we concluded that the PROMs overlap considerably.” By what measure/standard is the reported amount of overlap between the PROMs deemed ‘considerable’?
- Section 3.2, qualitative findings: There is no description of the qualitative themes in the results, which presumably are listed in the first column of Table 3.
- The qualitative quotes are incredibly insightful and, whilst recognising space limitations, it would be lovely to include a few brief/exemplary quotes throughout the results to further illustrate the findings.
- Did any findings arise in the interviews relating to QoL (introduced as a primary outcome in the introduction)?
- Was there any data collected on participants’ acceptability of, or feedback on, the program itself? If these data have been published previously, it would be useful to mention in the introduction.
Discussion:
- Lines 261-263: The following does not appear to be reported in the results and should either be added in, or removed from the discussion “The reductions in the participants’ total fatigue scores were remarkable, since no changes in their overall level of fatigue, as subjectively expressed by the participants themselves, had been reported over the preceding years.”
- Suggest moving the limitations toward the end of the discussion. The predominantly female sample should also be noted/discussed.
- Lines 278—292: This paragraph is better suited to the introduction.
- There is no discussion of the overlap of the PROMs noted in the results.
Figures/tables:
Figures: Please add explanations for the abbreviations used in all figures and tables in a footnote or write them in full if space allows.
Figure 2B: perhaps the y-axis is better labelled as “total depression score”, as described in the methods section.
Table 3: suggest adding column headings and number of participants who completed the interviews in the title of the table.
Author Response
[Responses from the Authors have been bolded]
Response to reviewer 2
We sincerely appreciate the favourable comments from the reviewer and the opportunity to submit a revised manuscript. We have, point by point, attempted to correct the manuscript accordingly.
Abstract:
Overall the abstract is a well written summary of the manuscript. There are some key details missing from the abstract however, which may have been presented in the “Simple summary” which should instead be focused on the background, aims and impact/importance of the research. I have highlighted these areas:
- The authors use cognitive [behavior or behavioural] therapy interchangeably when referring to CBT. Suggest using one term for consistency, preferrably the former. Done – used behavior systematically
- The abstract would benefit from a brief description of the Lightning Process program. Due to the word limit of abstract we cannot include brief description of the LP-programme. This, however, has already been briefly presented in the Simple Summary.
- Lines 31-32: It is not clear from the abstract what outcomes the semi-structured interviews assessed. Done, by expanding lines 33-34.
- Line 32: Please include the number of survivors recruited for the study in the results (which should be moved from the second background sentence) and basic demographic information to characterise the sample (e.g. age, sex, time since diagnosis if available). Done. However, Abstract word limit does not allow details on age, sex, time since diagnosis, which is presented in Table 1.
- Lines 33-34: “The correlation 33 coefficients between the various PROMs at baseline and the six-month follow-up point indicated 34 considerable overlap between the measures.” There is no detail provided about the “five validated patient-reported outcome measures” used, therefore these findings are not particularly valuable. It would be useful to briefly list them, or at least the overlapping outcome measures alluded to. Abstract word limit does not allow details here. That said, the five PROMs are very well known and widely used. The details on correlation coefficients are presented in the result section, page 9.
- Please add relevant statistics and/or p-values to the results. Done
- Lines 38-39: “These encouraging results here reported should be of interest to the general oncological community” Perhaps a more useful conclusion would be a summary of the key finding(s) or message or potential implications. As this is mentioned in detail in the Results just above, we hope it is acceptable to keep the conclusion as is.
- It is not clear from the abstract that the program is designed for young cancer survivors. Please add to the background or to the methods (program description or eligibility criteria). Done, see line 205-206 in the revised manuscript.
Introduction
The introduction offers a brief but adequate background of the topic, including engagement with relevant literature, to provide a good rationale for the study.
- The introduction is somewhat lacking in providing an understanding of the short/long term impact of (unaddressed) chronic fatigue in cancer survivors, and therefore the importance of programs like Lightning Process. Doing so would greatly strengthen the significance and rationale for the program and its evaluation. Done – see line 50-57.
- Line 63: Perhaps provide a brief definition of “myalgic encephalomyelitis” or more simply refer to it as “Chronic Fatigue Syndrome” We agree and have added “chronic fatigue syndrome” to comply with the most commonly used term. Besides from that, we would prefer not to go into more details about this controversial and complex diagnosis, as we consider this to falls outside the scope of our study and, as such, would take up unnecessary space.
- “This study sought to explore whether an intervention involving the LP could significantly reduce the level of fatigue and enhance the health-related quality of life (HR-QoL) 71 of AYAs who have been treated for sarcomas or lymphomas.” HRQoL is not mentioned as an outcome in the abstract and may be worth including if indeed it is a primary focus. Done – see line 27-28.
Methods:
The methods are generally well described and appropriate for the research aims, yet lacking some details.
- Lines 86-87: Was the program designed specifically for Lymphoma and Sarcoma patients, or was there another rationale for only recruiting these patients and not survivors of other diagnoses? Done – see lines 114-115. Could the program be relevant for other survivors or is further evaluation needed in these groups? We think it is reason to expect similar results in other cancer survivors, but as stated in the conclusion, we would await any clear recommendations until more robust clinical trials have been conducted.
- Lines 87-89: The following sentence should be moved to the results “The median time from diagnosis to the intervention was seven years, and the median time since receiving the final round of chemotherapy was six years (Table 1).” Table 1 and Table 2 should also be moved to the results. Instead, additional information about participant eligibility is needed, including any exclusion criteria. Done – see line 119-120.
- Lines 92-93: “The symptoms of chronic fatigue had repeatedly been documented in the participants’ clinical records as being their main late adverse effect following” Given that fatigue was recorded in patient records as a problematic late effect, could they have been receiving other treatment(s) to manage their fatigue? This is a valid point. With the timeline being 2-12 years since their primary diagnosis, some patients had indeed received other treatments to relieve their fatigue, but none of them with sustainable effects. This information was however not documented systematically as part of our study. Was this ruled out prior to participation, or noted, to account for the potential influence of other treatments which may have also improved fatigue and/or HRQoL? Since all participants in our study still had severe fatigue that had lasted for a long time prior to inclusion, potential influence of other treatments back in time was not likely.
- Lines 98-110: The description of the participants and their characteristics should appear at the start of the results, not in the methods. Done
- Line 122: Please define “NRH-OUH” in full. Done
- Lines 125-132: Additional information about the nature of the measures would be useful for the interpretation of the findings, i.e. possible response options and score ranges, scoring (e.g. higher score = more depression) Since all five PROMs are widely used/well known, and that the original scientific reference is cited in relation to each of the five containing information about the nature of the measures, we do not think further details on “response options, scoring ranges…..” is justified in our paper.
- Line 148: Please define “OUH-NRH” in full, if different to NRH-OUH Done
- Line 155: Additional information about the qualitative analysis is needed. Did LF and AM code unique transcripts, or an overlapping proportion? Both unigue and overlapping – see line 196. How was the coding tree developed and how were themes derived (in advance, or inductively)? Inductively – see line 194. What software (if any) was used to support analysis? We coded manually, as mentioned, and did not use any software in the analysis. What questions were asked in the interview and/or outcomes assessed? We have now added more information about this from line 178-185. If the editor/reviewers wish, we coudl translate the entire interview-guide to English and present it as a supplementary file.
- Lines 165-167: “and to allow the 165 instructor to assess whether they were sufficiently motivated to proceed with the intervention.” Without context, this may be interpreted as somewhat coercive. What measures were taken to ensure participants were ‘motivated’, or what criteria was used to determine if they were or were not? The LP-instructor did, during a telephone conversation, use a standard LP check-list as part of the interview about the participants motivations, see line 202- 204. During this interview, the participant’s readiness to change (in line with The Transtheoretical Model) is assessed. This is to ensure that only motivated and willing participants are included, and is as such more at the opposite end of coercive, as the aim is to ensure fully informed consent and proper selection.
Results:
Overall, the results are sound although could use some improvement in their presentation.
- Throughout the results, I suggest adding relevant statistics where results are described for completeness, for example the mean difference in scores, test statistics, and/or p-value. Done
- The readability/flow of the results would be enhanced by using the name of the outcome (e.g. fatigue, depression) rather than the acronyms of the formal measure names (e.g. FQ, PHQ), providing they are adequately described in the methods (as noted above). Alternatively, the results could benefit from additional subheadings for each outcome. Since the five PROMs are written in full and abbreviated in materials and methods, we suggest keeping too the abbreviations in the Result/later in the revised manuscript. Since PHQ does not only relate to depression it will be wrong just use the name of this outcome as suggested by the reviewer.
- Lines 189-191: “Any further improvements in terms of the….” This sentence isn’t clear. Perhaps reword to indicate more clearly that a reduction in fatigue was also observed in the remaining three patients, although these changes were not significant. Done. See lines 233-235
- Lines 205-206: A brief summary of the PCS findings are needed, rather than simply stating “The analysis concerning the PCS is shown in Supplemental Table 5.” We have added text on lines 252-253: But here there were no significant improvement documented.
- Lines 207-208: Please move the following to the methods: “The CSQ-8 was completed one week after the participants finished the LP course and 207 again at the six-month follow-up point…higher scores indicat[ed] a higher degree of satisfaction.” Done. Pasted inn lines 144-147. It would also be useful to know what the total possible score is, to understand the scores better. Done. We have expanded the text also mentioning 32 as the total possible best score. See lines 156-158.
- Lines 214-220: “A similar calculation comparing the participants’ PHQ-9 and FQ score…” This is vague. How is it different to the data presented a few sentences prior “A moderate overlap 214 between the participants’ PHQ-9 and FQ scores was observed at baseline…”. Done. We have clarified the various timepoints being both at baseline and six months follow up. See lines 263-270.
- Lines 221-222: “Hence, we concluded that the PROMs overlap considerably.” By what measure/standard is the reported amount of overlap between the PROMs deemed ‘considerable’? Done. See added sentence in 2.2.2 Statistical analyses.
- Section 3.2, qualitative findings: There is no description of the qualitative themes in the results, which presumably are listed in the first column of Table 3. Done see lines 273-279
- The qualitative quotes are incredibly insightful and, whilst recognising space limitations, it would be lovely to include a few brief/exemplary quotes throughout the results to further illustrate the findings. Done, incorporated four quotations, see lines 286-319.
- Did any findings arise in the interviews relating to QoL (introduced as a primary outcome in the introduction)? During the interview, participants were not explicitly asked about QoL, but what they expressed as improved well-being physical and psychological functioning, better family life, social life, vocational life and were taken as improved quality of life – see line 178-185.
- Was there any data collected on participants’ acceptability of, or feedback on, the program itself? If these data have been published previously, it would be useful to mention in the introduction. No data from this study has previously been published, and this is the first study of LP applied to cancer-related fatigue. Although acceptability of the program was not explicitly assessed in the study, we consider the qualitative data as well as high adherence to indirectly support acceptability of the program.
Discussion:
- Lines 261-263: The following does not appear to be reported in the results and should either be added in, or removed from the discussion “The reductions in the participants’ total fatigue scores were remarkable, since no changes in their overall level of fatigue, as subjectively expressed by the participants themselves, had been reported over the preceding years.” The positive improvement in total fatigue scores has, indeed, been presented in Results (with P-values). See lines 228-232.
- Suggest moving the limitations toward the end of the discussion. Done see lines 412-421. The predominantly female sample should also be noted/discussed. Done, noted in lines 413.
- Lines 278—292: This paragraph is better suited to the introduction. Done. Part of the paragraph is moved to introduction, line 73-79, but slightly condensed to avoid repeating text above.
- There is no discussion of the overlap of the PROMs noted in the results. Due to the low number of participants in this study we do not deem it justified to further elaborate on this quite complex matter in the Discussion (just keep it, as now presented and slightly expanded in Results). See lines 261-271.
Figures/tables:
Figures: Please add explanations for the abbreviations used in all figures and tables in a footnote or write them in full if space allows. Done
Figure 2B: perhaps the y-axis is better labelled as “total depression score”, as described in the methods section. We agree, and have adjusted the y-axis as suggested and corrected accordingly in text line 152.
Table 3: suggest adding column headings and number of participants who completed the interviews in the title of the table. Done
